# Gating interactions steer loop conformational changes in the active site of the L1 metallo-β-lactamase

Zhuoran Zhao[1†], Xiayu Shen[1†], Shuang Chen[1], Jing Gu[1], Haun Wang[1], Maria F Mojica[2,3,4], Moumita Samanta[5], Debsindhu Bhowmik[6], Alejandro J Vila[4,7,8], Robert A Bonomo[2,3,4,9], Shozeb Haider[1,10]*

[1]Department of Pharmaceutical and Biological Chemistry, School of Pharmacy, University College London, London, United Kingdom; [2]Department of Molecular Biology and Microbiology, Case Western Reserve University School of Medicine, Cleveland, United States; [3]Louis Stokes Cleveland Department of Veterans Affairs Medical Center, Cleveland, United States; [4]CWRU-Cleveland VAMC Center for Antimicrobial Resistance and Epidemiology (Case VA CARES), Cleveland, United States; [5]College of Computing, Georgia Institute of Technology, Atlanta, United States; [6]Computer Science and Engineering Division, Oak Ridge National Laboratories, Oak Ridge, United States; [7]Laboratorio de Metaloproteínas, Instituto de Biología Molecular y Celular de Rosario (IBR, CONICET-UNR), Rosario, Argentina; [8]Area Biofísica, Facultad de Ciencias Bioquímicas y Farmacéuticas, Universidad Nacional de Rosario, Rosario, Argentina; [9]Departments of Medicine, Biochemistry, Pharmacology, and Proteomics and Bioinformatics, Case Western Reserve University School of Medicine, Cleveland, United States; [10]UCL Centre for Advanced Research Computing, University College London, London, United Kingdom

*For correspondence:
shozeb.haider@ucl.ac.uk

†These authors contributed equally to this work

**Abstract** β-Lactam antibiotics are the most important and widely used antibacterial agents across the world. However, the widespread dissemination of β-lactamases among pathogenic bacteria limits the efficacy of β-lactam antibiotics. This has created a major public health crisis. The use of β-lactamase inhibitors has proven useful in restoring the activity of β-lactam antibiotics, yet, effective clinically approved inhibitors against class B metallo-β-lactamases are not available. L1, a class B3 enzyme expressed by *Stenotrophomonas maltophilia*, is a significant contributor to the β-lactam resistance displayed by this opportunistic pathogen. Structurally, L1 is a tetramer with two elongated loops, α3-β7 and β12-α5, present around the active site of each monomer. Residues in these two loops influence substrate/inhibitor binding. To study how the conformational changes of the elongated loops affect the active site in each monomer, enhanced sampling molecular dynamics simulations were performed, Markov State Models were built, and convolutional variational autoencoder-based deep learning was applied. The key identified residues (D150a, H151, P225, Y227, and R236) were mutated and the activity of the generated L1 variants was evaluated in cell-based experiments. The results demonstrate that there are extremely significant gating interactions between α3-β7 and β12-α5 loops. Taken together, the gating interactions with the conformational changes of the key residues play an important role in the structural remodeling of the active site. These observations offer insights into the potential for novel drug development exploiting these gating interactions.

## Editor's evaluation

In this useful study, the authors utilize state-of-the-art computational methods complemented with some experimental validation to investigate the dynamics of flexible loops of the L1 Metallo-β-lactamase enzyme, resulting in a better understanding of the various conformational states useful for the rational design of superior β-lactamase inhibitors/antibiotics. The evidence supporting the claims is solid, and the work will be of interest to computational, experimental biologists, and drug designers working on antibiotic resistance.

## Introduction

β-Lactam antibiotics are the most important and widely used class of antibiotics for treating bacterial infections (*Bush and Bradford, 2016*). They exhibit high efficacy and minimal side effects. Penicillins, carbapenems, cephalosporins, and monobactams are typical β-lactam antibiotics on the market and, accounted for 65% of all antibiotics prescriptions in the United States (*Bush and Bradford, 2016*). A common structural feature in this exceptionally large class of antibiotics is the presence of a β-lactam ring, which acts as a false substrate for the transpeptidases (also called Penicillin-binding proteins, PBPs) and inhibits peptidoglycan cross-linking, thereby impairing the synthesis of the bacterial cell wall (*Palzkill, 2018*). While all β-lactam antibiotics share a common mode of action, they however, exhibit distinct properties in terms of spectrum of action, pharmacokinetics and, to some extent, activity against resistant strains (*Van Bambeke et al., 2017*).

Recognizing that β-lactams have shown great benefits in clinical treatments, the overuse and misuse of β-lactams have favored the emergence of β-lactam resistance, raising public concerns (*Tooke et al., 2019*). As the resistance has progressed, β-lactams have experienced a rapid potency loss against various bacteria (*Tooke et al., 2019*). The World Health Organization projected in 2019 that such drug resistance could cause up to 10 million deaths annually by 2050 ('No time to Wait', n.d.). Therefore, dealing with β-lactam antibiotics resistance is an urgent global public health challenge.

Three established mechanisms through which bacteria develops resistance against β-lactam antibiotics are previously known (*Mora-Ochomogo and Lohans, 2021*). These include but not limited to the production of β-lactamase enzymes, modification of the target protein – PBPs, or preventing β-lactam antibiotics from reaching the target. Among these, the expression of β-lactamases is the most common mechanism found in various bacteria resistant to β-lactams (*Lingzhi et al., 2018*). β-Lactamases inactivate β-lactam antibiotics by hydrolyzing the amide bond in the β-lactam ring.

Four classes of β-lactamases are established. While classes A, C, and D are serine β-lactamases, class B groups the zinc-dependent metallo-β-lactamases (MBLs). MBLs employ one or two zinc ions at the active site as catalytic cofactors (*Palzkill, 2013*; *Tooke et al., 2019*). MBLs are further subdivided into three subclasses (B1, B2, and B3) based on their sequence, metal content, and different active site features (*Bahr et al., 2022*; *Bahr et al., 2021*). All MBLs adopt the αβ/βα sandwich motif and have mono- (class B2) or binucleated zinc (classes B1 and B3) at the active site. The B1 and B3 subclasses have a broad-spectrum substrate profile that includes penicillins, cephalosporins, and carbapenems, while the B2 enzymes exhibit a narrow profile that includes carbapenems (*Palzkill, 2013*; *Tooke et al., 2019*). Such distinctive ability to hydrolyze the new generation of carbapenem antibiotics, the rapid emergence of new clinical variants, the potential for interspecies transfer in the presence of mobile genetic elements, isolation from nosocomial and environment sources, coupled with the unavailability of any clinically useful MBL inhibitors makes MBLs of significant clinical interest and concern (*Bush and Bradford, 2020*; *Mojica et al., 2022*; *Tooke et al., 2019*).

*Stenotrophomonas maltophilia* is a non-fermenting, Gram-negative bacillus that has emerged as an opportunistic pathogen. This organism mainly affects patients with cancer, cystic fibrosis, and other conditions leading to an immunosuppressed state, where it can cause a variety of clinical syndromes, mainly pulmonary and blood stream infections (*Mojica et al., 2019*). Crude mortality rates range from 14% to 69% in patients with bloodstream infection due to *S. maltophilia* (*Adegoke et al., 2017*; *Brooke et al., 2017*; *Crisp et al., 2007*). β-Lactams resistance in *S. maltophilia* is mediated by the coordinated expression of L1 metallo-β-lactamase (class B3 MBL) together with L2 β-lactamase (class A β-lactamase) (*Okazaki and Avison, 2008*). L1 MBL can hydrolyze almost all β-lactam antibiotics with the exception of monobactams such as aztreonam and is resistant to all clinically available β-lactamase inhibitors (*Brooke, 2012*; *Mojica et al., 2019*; *Okazaki and Avison, 2008*).

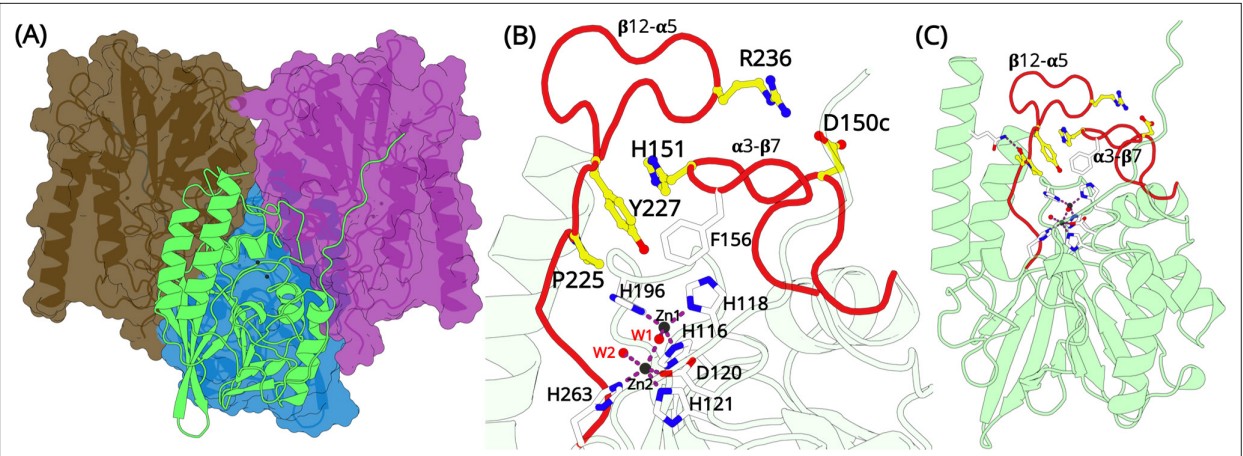

**Figure 1.** Overall structural features of the L1 metallo-β-lactamase (MBL). (**A**) The homo-tetrameric conformation of L1 MBL (PDB entry 1SML). Three monomers are illustrated in surface representation; the fourth monomer is shown in green cartoon. (**B**) A close up of the zinc-binding site and the elongated α3-β7 and β12-α5 loops (red). Two zinc ions are present at the binding site. Zn1 coordinates with residues H116, H118, H196, and a water molecule forming a tetrahedral geometry. Zn2 is trigonal bipyramidal and coordinates with residues D120, H121, and H263. Zn1 and Zn2 are bridged by a water molecule. The position of D150c, H151, P225, Y227, and R236 are shown as yellow sticks. (**C**) L1 monomer illustrated as cartoon (green); the position of the two elongated loops (red) that forms a part of the active site in each monomer are highlighted in red.

The online version of this article includes the following figure supplement(s) for figure 1:

**Figure supplement 1.** Catalytic cycle of L1 metallo-β-lactamase (MBL).

**Figure supplement 2.** Sequence alignment of seven class B metallo-β-lactamase with the BBL numbering.

L1 MBL is unique among all known β-lactamases as it is the only known β-lactamase that functions as a tetramer (*Figure 1A*). In each subunit, there are two zinc ions bound at the active site (*Figure 1B*, *Ullah et al., 1998*). The first Zinc (Zn1) is in a tetrahedral coordination with a water molecule and three histidine residues. The second Zinc (Zn2) is coordinated by two histidine, one aspartic acid and a water molecule in a trigonal bipyramidal geometry. Another water molecule, which acts like a nucleophile for the catalytic reaction bridges the two zinc ions (*Figure 1—figure supplement 1*). Studies have revealed that both zinc ions are relevant for hydrolysis and protein folding in L1 (*Crisp et al., 2007*; *Hu et al., 2008*; *Ullah et al., 1998*).

The L1 MBL tetramer is held tightly by three discrete sets of hydrophobic interactions, which allows each monomer to make contacts with the other three subunits (*Crisp et al., 2007*; *Sevaille et al., 2017*; *Ullah et al., 1998*). All these interactions are primarily hydrophobic and lead to each monomer burying 1300 Å² of non-polar side-chain surface area on the formation of the tetramer. A two-fold inverted symmetry of the subunit arrangement positions some important residues and interactions at the subunit interfaces in close proximity (*Figure 1A*). The key interactions made by the side chain of M175 penetrating a shallow pocket formed by E141, L150d, P198, and Y236 (*Figure 1—figure supplement 2*; after the standard BBL numbering *Garau et al., 2004*) on the surface of the adjacent subunit, has been shown to be essential for stabilizing the tetramer (*Simm et al., 2002*). A substitution of this M175 results in failure of tetramerization and a reduced activity against most β-lactams of ca. two orders of magnitude (*Simm et al., 2002*). How does the formation of a tetramer benefits L1 MBL to confer resistance remains unknown.

One distinct feature of L1 MBL is the presence of two elongated loops (α3-β7 and β12-α5 loops; *Table 1*) surrounding the metal center (*Figure 1B, C*). The α3-β7 loop at the N-terminal extension has 20 residues (G149-D171) and extends over the active site (*Garrity et al., 2004*). These residues are considered to be involved in determining the spectrum of the enzyme by forming hydrophobic

**Table 1.** Elongated loops.

| α3-β7 | GGSDDLHFGDGITYPPANAD Residues 149–171 |
|---|---|
| β12-α5 | ADSLSAPGYQLQGNPRYPH Residues 219–239 |

interactions with large hydrophobic motifs at C2 or C6 of β-lactam substrates (*Sevaille et al., 2017*; *Spencer et al., 2005*; *Spencer et al., 2001*; *Ullah et al., 1998*). Two novel substitutions were identified in vitro, on the α3-β7 loop: G159A and (G/E)161D. The role of G159 in substrate binding and catalysis has been determined by the substitution of G159A. This substitution disrupts the positioning of F156, another significant residue involved in the function of L1, by decreasing the flexibility of the loop. Similarly, the G161D substitution can also affect the efficiency of the enzyme catalytic activity (*Carenbauer et al., 2002*; *Nauton et al., 2008*; *Spencer et al., 2005*; *Ullah et al., 1998*).

The β12-α5 loop consists of 19 residues (A219-H239) and lies adjacent to the active site. S221 in this loop makes indirect interaction with Zn2, while S223 serves as a second-shell residue of H196 (*Crisp et al., 2007*; *Mojica et al., 2019*; *Ullah et al., 1998*). S221 coordinates a Zn2-bound apical water, whereas S223 directly interacts with the substrate C4 carboxylate group (*Spencer et al., 2005*). In a recent crystal structure, S221 and S223 have been found to directly interact with substrates (*Twidale et al., 2021*). Besides, novel substitutions have been identified in the loop, namely Q230R, (K/Q)232R, G233D, and (P/A)235V (*Mojica et al., 2019*). Although none of these residues seems to directly interact with the substrate, it is likely that these substitutions will affect the catalytic activity either by strengthening or weakening the H-bonding network that maintains the loop in place or by changing its electrostatic characteristics (*Tooke et al., 2019*). The apical points of α3-β7 and β12-α5 loops are positioned at the interface between monomers, and direct interaction with residues that affects the catalytic activity of the L1 MBL has been confirmed (*Ullah et al., 1998*).

The crystal structure of L1 MBL has been determined in apo state and in complex with inhibitors and final reaction products (*Nauton et al., 2008*; *Sevaille et al., 2017*; *Twidale et al., 2021*; *Ullah et al., 1998*). In all structures, the substrate/inhibitor-binding site (active site from here onwards) is well defined. In the apo state, the β12-α5 loop is collapsed on the active site. In complex structures, the β12-α5 loop wraps around to enclose the substrate/inhibitor within the active site. It is worth noting that the conformation of the α3-β7 and β12-α5 loops are identical in all structures, be it in the apo state or in complex with substrate/inhibitors. In this collapsed, closed conformation, residues R236 (β12-α5) and D150c (α3-β7) form a salt bridge interaction; Y227 (β12-α5) forms a π–π stacking interaction with H151 (α3-β7) and P225 (β12-α5) adopts an 'in' conformation, where the Cγ atom in the cyclic side chain of P225 is 6.5 Å from the centroid of the two Zinc ions (*Figure 1B*). An additional interaction between the side chain of the Q310 (α6 helix) and the backbone carbonyl oxygen of A224 (β12-α5) stabilizes the loops in the closed conformation (*Figure 1C*). Previous studies have shown that the inter-subunit interactions are the major contributors that affect the flexibility of α3-β7 and β12-α5 loops, and the dynamics of these loops can directly influence β-lactam binding and catalysis (*Spencer et al., 2005*). These findings have indicated the importance of inter-subunit interactions in inhibitor design. While the catalytic mechanism of β-lactam hydrolysis in MBLs is well established, the mechanistic insights into how the dynamics of these flexible loops affect the structural rearrangements of the enzyme active site remains unknown.

To sample the conformational landscape explored by L1 MBL, we performed adaptive sampling simulations (cumulative sampling over 100 μs) and investigated the metastability of the α3-β7 and β12-α5 loops via Markov State Models (MSMs). To visualize the major metastable conformations of the loops alone, unsupervised low-dimensional embeddings were created using a convolutional variational autoencoder (CVAE). Observations from MSMs and CVAE-based deep learning identified important conformational changes in the α3-β7 and β12-α5 loops, which have significant effects on the dynamic architecture of the enzyme active site and may directly affect the catalytic activity of L1 MBL. Five residues, D150c, H151 (α3-β7 loop) and P225, Y227 and R236 (β12-α5 loop) were substituted and the activity of the engineered L1 variants was evaluated in cell-based experiments. Our results provide strong evidence that the dynamics of α3-β7 and β12-α5 loops are extremely critical for the proper biological function of L1 MBL.

## Results and discussion
### Markov state models

A converged MSM with seven metastable states was successfully built using the backbone torsions and the Cα-RMSD of the α3-β7 and β12-α5 loops, at a lag time of 5 ns (*Figure 2A*). The selected features resolved the Markovian model and enabled the analysis of the representative conformational

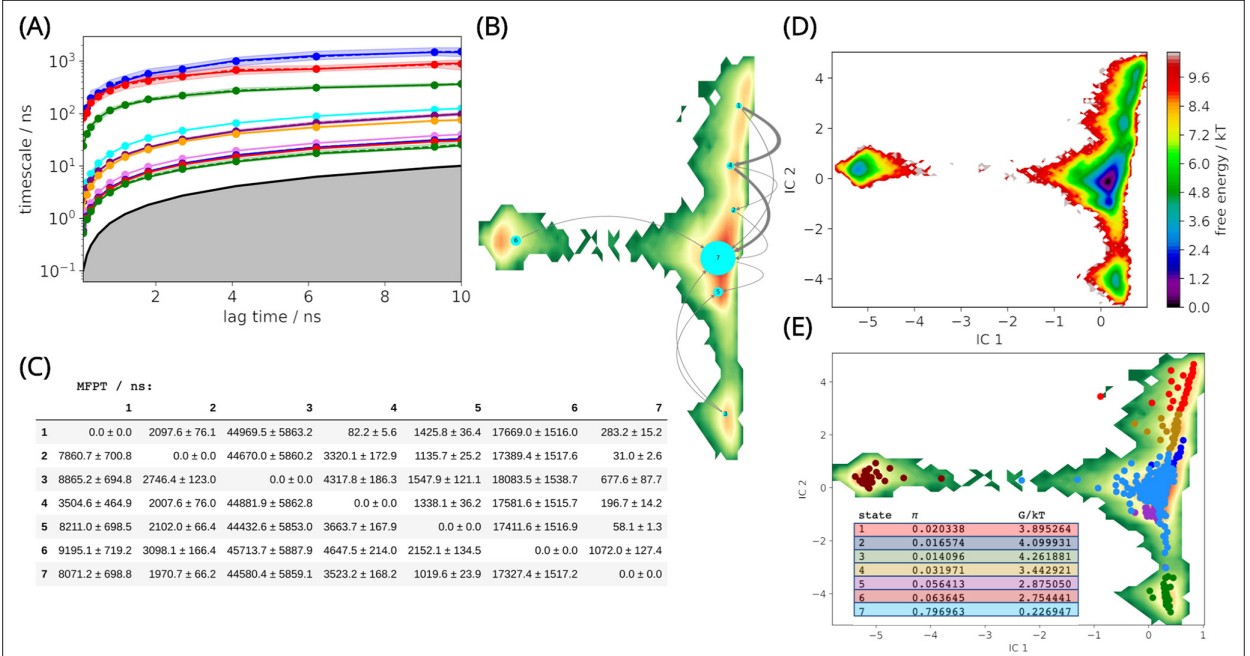

**Figure 2.** L1 metallo-β-lactamase Markov State Model. (**A**) Implied timescales (ITS) plot. The ITS plot describes the convergence behavior of the implied timescales associated with the processes at the lag time of 5 ns. (**B**) Net flux pathways plot highlights the transition pathways in the relevant directions, between the highest energy state 1, and all other states. The thickness of the arrows between states suggests the possibility of each transition. The thicker the arrow, the greater the possibility. (**C**) Mean first passage times between metastable states. (**D**) Free energy landscape. The sampled free energy projected onto the first two time-lagged independent components (ICs) at lag time $\tau$ = 5.0 ns. (**E**) The distribution of cluster centers highlighting the presence of the metastable states. Macrostate distributions of conformations projected onto the first two ICs identified seven macrostates. The population of each state ($\pi$) and its free energy estimation is listed.

The online version of this article includes the following source data and figure supplement(s) for figure 2:

**Figure supplement 1.** Chapman–Kolmogorov (CK) test plot.

**Source data 1.** Mean first passage time.

changes that occur when transitioning between different states (*Figure 2—figure supplement 1*). Furthermore, the chosen features permitted us to make valid links between the free energy (FE) landscape and the dynamics of the α3-β7 and β12-α5 loops. From the MSM results, significant structural differences can be observed by comparing the conformations obtained from different metastable states.

## The α3-β7 and β12-α5 loops exist in open, intermediate, and closed states

The dynamics of the two loops (α3-β7 and β12-α5) can be described by closed, intermediate, and open conformations (*Figure 3—figure supplement 1*). The 'closed' conformations of the loop is defined by the collective formation of a salt bridge between D150c-R236 (α3-β7 and β12-α5), H151-Y227 π–π stacking interaction and P225 adopting an 'in' conformation (*Figure 3A*; *Figure 3—figure supplements 2–4*). The 'in' conformation is defined as when the Cγ atom of P225 is oriented toward the zinc atoms in the active site. The distance between the Cγ atom and the centroid of the two zinc atoms is calculated to be 6.5 Å in the 1SML crystal structure (*Ullah et al., 1998*). There is an additional interaction between the side chain of Q310 (α6 helix) and the backbone of A224 (β12-α5). This interaction tethers the terminal α6 helix to the rest of subunit and stabilizes the β12-α5 loop (*Figure 3—figure supplement 5*). The loops are collapsed on the active site resulting in an occluded 420.4 ± 94.8 Å³ cavity (*Figure 3A*). The structure of the active site is analogous to the crystalline conformation, thereby highlighting it as a relevant functional state. State 7 defines the closed conformation of L1 MBL, with the Cα-RMSD comparison with the apo state crystal structure (PDB id 1SML) to be 0.79 Å and with inhibitor complex (PDB id 5DPX) is

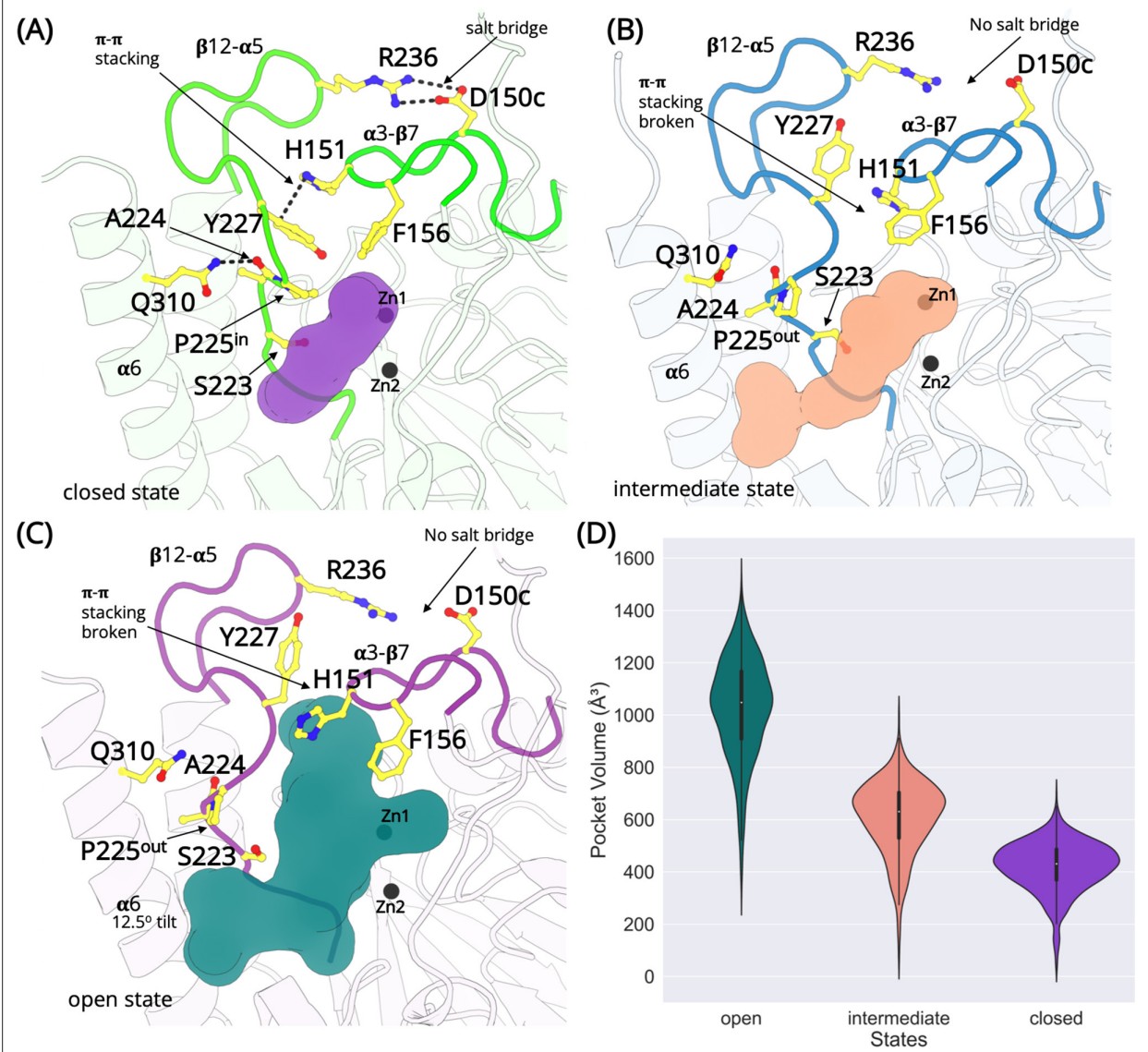

**Figure 3.** The L1 metallo-β-lactamase active site remodeling. Structures exist in the closed, intermediate, and open states defined by a salt bridge between D150c-R236, π–π stacking interaction between H151-Y227 and the conformation of the cyclic side chain of P225. (**A**) In the closed state, the salt bridge and the π–π stacking interaction is formed and P225 adopts the 'in' conformation. The β12-α5 loop is collapsed occluding the active site (purple). (**B**) In the intermediate open state, the interactions are lost and P225 adopts the 'out' conformation. However, the β12-α5 loop is not completely in the open conformation. A small cavity between α6 helix and β12-α5 is formed that merges with the active site (pink). (**C**) In the fully open state, the salt bridge and the π–π stacking interaction is lost. P225 adopt 'the out' conformation. The β12-α5 loop moves outwards, opening a large cavity that merges with the active site (teal). (**D**) The calculated volume of the active site in different states.

The online version of this article includes the following figure supplement(s) for figure 3:

**Figure supplement 1.** Probability weighted distance distributions in the open, intermediate, and the closed states.

**Figure supplement 2.** Salt bridge interaction between D150c-R236.

**Figure supplement 3.** π–π stacking interaction between H151-Y227.

**Figure supplement 4.** Distance between P225 and the centroid of zinc ions.

**Figure supplement 5.** Interaction between A224-Q310.

**Figure supplement 6.** Structural comparison of the L1 metallo-β-lactamases.

0.77 Å (*Figure 3—figure supplement 6*). State 7 is the most populated, with 80% of all conformations falling within this cluster. State 7 is also the lowest energy state. These observations allow us to conclude that the closed state dominates the dynamics and also explains why this conformation is always resolved in crystallographic experiments. States 3, 5, and 6 are sub-states of the closed conformation. In addition to the slight differences in the conformations of α3-β7 and β12-α5 loops, minor changes are also observed in the conformations of the adjacent loops, at the distal end of the active site.

State 2 is an intermediate closed state where the H151-Y227 π–π stacking interaction is formed and P225 adopts an 'in' conformation. However, the D150c-R236 salt bridge is not fully formed. State 4 is an intermediate open conformation (*Figure 3B*), where two or more interactions including the D150c-R236 salt bridge, H151-Y227 π–π stacking, are lost and P225 adopts the 'out' conformation. The outward movement of the β12-α5 loop results in the interaction between the side chain of Q310 (α6 helix) and the backbone carbonyl oxygen of A224 (β12-α5) to break, causing the α6 helix to tilt away from the binding site. This generates a small cavity between α6 helix and the β12-α5 loop, which merges with the active site. The average volume calculated in the intermediate state is 608.3 ± 137.25 Å$^3$.

In the open conformation (state 1), the D150c-R236 salt bridge interaction is broken, the H151-Y227 π–π stacking interaction is lost and the cyclic side chain of P225 orients away from the active site and adopts an 'out' conformation. A loss of these interactions allows the β12-α5 loop to move away from the active site. Additionally, this structural rearrangement leads to a complete loss of the tethering interaction between the side chain of Q310, positioned on the C-terminal α6 helix and the backbone of A224, causing the α6 helix to tilt by 12.5° away from the active site. The loss of constraints around the loop allows the side chain of Y227 to rotate, with the hydroxyl group orienting toward R236 and away from the active site. The loss of the H151-Y227 π–π stacking interactions and the rotation of the Y227 side chain exposes an additional cavity that merges with the pocket identified in the intermediate open state. The total volume of the merged cavity is 1027.17 ± 191.04 Å$^3$ and is large enough for the substrates to access the zinc metal center.

The net flux pathways between states and the probabilities of each pathway have been estimated (*Figure 2B*). The shortest mean first passage time in the flux pathways is between states 1 → 4 → 7, which indicates that the free energy (FE) barrier along this path is low (*Figure 2C*). This can explain why this flux pathway shows predominant prevalence (89.7%) at equilibrium. This flux pathway is followed by 1 → 7 (9.1%) and 1 → 4 → 2 → 7 (1.2%). The long mean first passage times to visit states 3, 6 from 1 indicate that the FE barrier is high (*Figure 2C, D*).

Based on the results, the conformational changes that occurred between state 1, 4, and 7 supports the best explanation to the dynamics of the L1 MBL, as over 89% of the conformations transit from state 1 to state 7 via state 4 (*Figure 2B*), and any important conformational changes occurring around the active site could be related to the conformations of the two loops (α3-β7 and β12-α5 loops) identified in these states.

## Deep learning analysis of α3-β7 and β12-α5 interactions

L1 MBL, like all other β-lactamases, comprises of a hydrophobic core, that is stabilized by packing interactions that prevent long distance motions during simulations (*Galdadas et al., 2021*; *Galdadas et al., 2018*; *Olehnovics et al., 2021*). This indicates that thermal vibrations at fast timescales may significantly obscure potentially important slow dynamics. Furthermore, differentiating important interactions and correlations with classified sub-states in a 1064 residue homotetrameric L1 MBL proved non-trivial. This was additionally compounded by >1 million data points representing sampled conformations.

To resolve this, an unsupervised CVAE-based deep learning was performed (*Bhowmik et al., 2018*; *Romero et al., 2019*). The CVAE provides biophysically relevant information related to conformational transitions induced by changes in interactions. More specifically, CVAE was used to identify dynamic changes in the α3-β7 and β12-α5 loops resulting from changes in D150c-R236 salt bridge, H151-Y227 π–π stacking interaction and the 'in/out' conformation of P225.

A 39 × 39 residue symmetric distance matrix, representing the Cα residues of α3-β7 (G149-D171) and β12-α5 loops (A219-H239), was used to featurize the dynamics of the loops. A previously tested

protocol comprising of 64 filters in each of the 4 convolutional layers, and no pooling layers maintained a deep representational space of trainable parameters in each hidden layer (*Olehnovics et al., 2021*).

To enhance the learning quality of the CVAE model, conformations from each state were mixed together followed by the evaluation of the training and validation loss of the combined dataset. The CVAE model was implemented and tested from dimensions 3 to 30. As the dimension size increases, the model compresses less and exhibits more representation capability. When the latent dimension becomes too large, the model may overfit local features and introduce extra noise. The optimal value of the overall loss is somewhere between these two extremes. For the current dataset, the CVAE model is quite stable and robust, as highlighted by the low deviation of validation loss across different latent dimensions (*Figure 4—figure supplement 1*).

The reconstruction loss was calculated and the final latent dimension of 14, representing the lowest dimension loss value was selected. Following this, the t-distributed stochastic neighbor embedding (tSNE) was performed on the compressed reduced dimension data for visualization in two dimensions

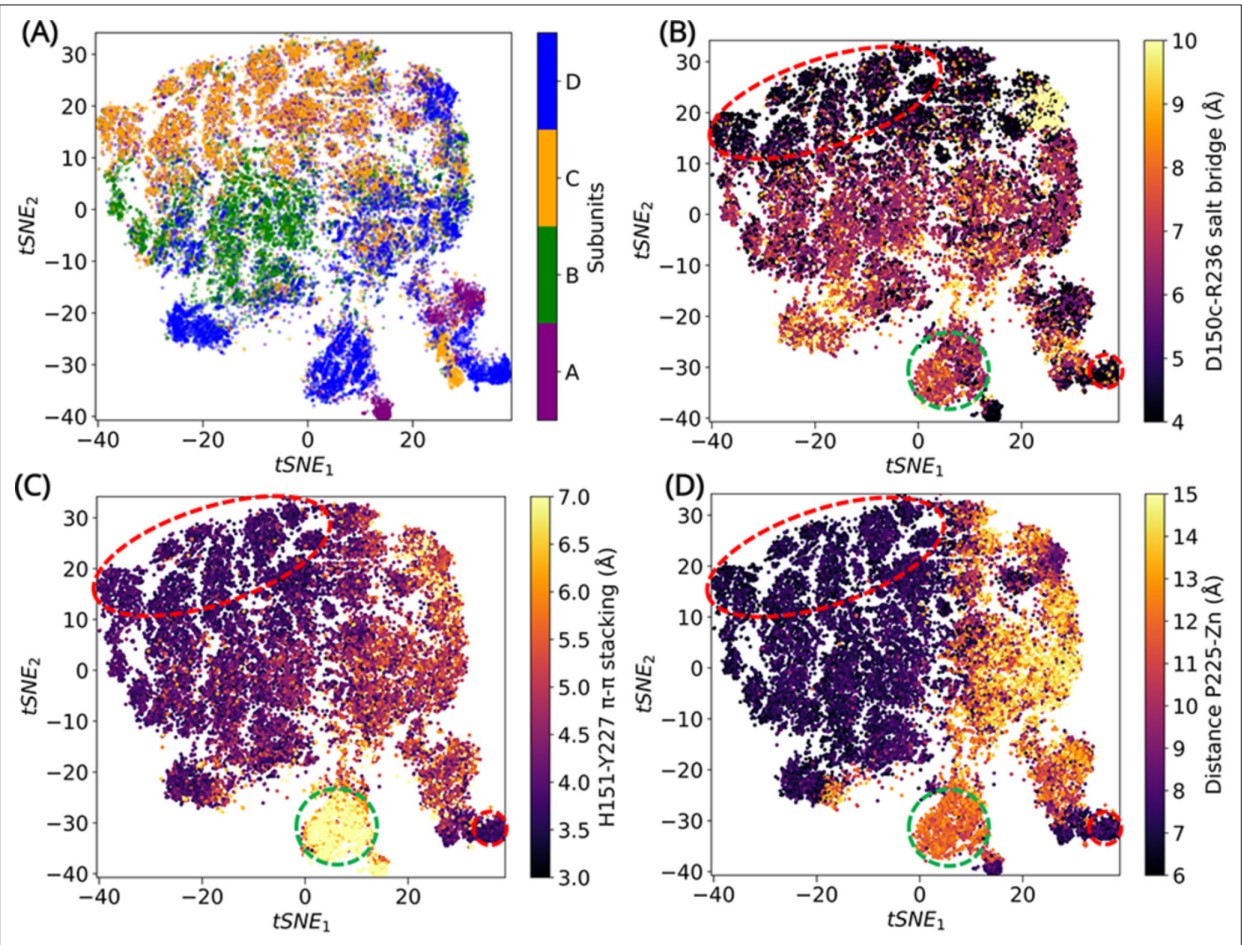

**Figure 4.** Convolutional variational autoencoder (CVAE)-based deep learning analysis. CVAE-learned features of high-dimensional data represented in 2D following t-distributed stochastic neighbor embedding (tSNE) treatment. (**A**) The results show that A and C subunits show similar dynamics, while B and D subunits cluster together. The (**B**) salt bridge interactions between D150c-R236, (**C**) H151-Y227 π–π stacking interaction, and (**D**) distance between P225 and the centroid of Zn atoms have been plotted. The red box highlights a region where closed conformations are found, while the green box highlights a region of open conformation. The other conformations are open intermediate (two or more features are lost) or closed intermediate (two or more features are present).

The online version of this article includes the following figure supplement(s) for figure 4:

**Figure supplement 1.** Convolutional variational autoencoder (CVAE)-based deep learning analysis.

**Figure supplement 2.** Clustering metastable states via t-distributed stochastic neighbor embedding (tSNE).

(*Maaten and Hinton, 2008*). The combined deep learning is able to cluster individual subunits in the tetramer based on their local and global conformational dynamics (*Figure 4—figure supplement 2*).

The overall dynamics representing various conformations sampled by each monomer can be identified using the tSNE visualization of the CVAE latent space. These conformations are resolved based on the dynamics generated by the distance matrix of the α3-β7 and β12-α5 loops. While all subunits sample similar motions, the conformations are clustered based on comparable dynamics. Subunits A and C share similar dynamics, while subunits B and D are clustered in the same tSNE space (*Figure 4A*). The salt bridge, π–π stacking interactions and P225 'in/out' conformations are all observed, although the closed and intermediate conformations dominate the states. Subunit B displays low flexibility, with most conformations unable to form a complete salt bridge interaction. In spite of this, a complete π–π stacking interaction and the P225 'in' conformation is observed in a low population of B-subunits. The most flexible subunit is D, where all three, 'open–intermediate–closed' states are observed.

To confirm that the states indeed describe the 'open' and 'closed' conformations of the α3-β7 and β12-α5 loops, three features were measured – the D150c-R236 salt bridge interaction (*Figure 4B*); H151-Y227 π–π stacking interaction (*Figure 4C*) and the distance between P225 and the centroid of Zn atoms (*Figure 4D*). The features were then compared by projecting 1000 conformations extracted from MSM-derived metastable states, on to the tSNE space (*Figure 4—figure supplement 2*). This permitted visualization of exemplar conformations that best represented open–intermediate–closed states. The conformations sampled in metastable states 1 and 4 are similar. While the dynamics can be described by more than one representation, subunit D in state 1 describes the open state, subunit A in state 4 represents the intermediate open conformation, subunit B in state 2 describes intermediate closed conformation, and subunit C in state 7 was chosen to represent the closed conformation of L1 MBL.

## The conformational dynamics of the α3-β7 and β12-α5 loops

The salt bridge formation between D150c-R236 (α3-β7 and β12-α5) is crucial for the function of L1 MBL (*Simm et al., 2002*). This salt bridge acts as a gate, and the formation/loss of the salt bridge regulates the highly mobile β12-α5 loop to adopt an open/closed conformation. This configuration of the loop is driven by the conformation of the R236 side chain in the β12-α5 loop as it changes considerably while that of α3-β7, where D150c is positioned, remains rigid (*Figure 3*). Previous experiments on TEM-1, a class A β-lactamase, offered evidence for the significance of salt bridge formation between important loops (*Vakulenko et al., 1995*). Disruption of the salt bridge between R164 and D179 would in turn alter the structure of the substrate-binding site by restructuring the conformation of the Ω-loop, decreasing the antibiotic resistance (*Vakulenko et al., 1995*). Therefore, similarly for L1 MBL, it can be inferred that the formation of the salt bridge observed in the 'closed' conformation of the loops is critical in stabilizing the topology of the active site and enhancing the antibiotic resistance.

The π–π stacking interaction between H151 and Y227 is another important interaction that stabilizes the loops. In the closed state (loop 'closed'), the π–π stacking interaction is stable. However, the loss of the D150c-R236 salt bridge interaction generates enough space to permit the rotation of the side chain of Y227 from pointing to the zinc atoms in the active site to orienting toward R236, resulting in the loss of the π–π stacking interaction (*Figure 3*). This formation and loss of π–π stacking might be influenced by the formation of the salt bridge at the apical tips of the α3-β7 and β12-α5 loops. Such π–π stacking interaction is considered a very important interaction in protein folding and molecular recognition (*Brylinski, 2018*). Studies on other β-lactamases have revealed that the π–π stacking near the substrate-binding site can help stabilize the binding site topology and contribute to the recognition of the antibiotics (*Hargis et al., 2014*; *Langan et al., 2016*).

The side chain of P225 exists in trans configuration in L1 MBL, which is very unusual among the β-lactamases. Proline exists in two configurations in nature – cis and trans. Such configurations are defined by the ω torsion angle of proline (cis: close to 0°, trans: nearly 180°). Usually, in β-lactamases, the proline is found in the cis configuration. Studies on TEM-1 β-lactamase have already revealed the importance of the proline cis configuration in the protein folding and correct orientation of residues involved in the catalytic reaction (*Vanhove et al., 1996*; *Vanhove et al., 1995*). Mutations at this proline, near the binding site prevents the enzyme from functioning properly (*Vanhove et al., 1996*). However, in L1 MBL, P225 is found in trans configuration. To confirm this, the ω torsion angle was calculated throughout the course of the simulation (*Figure 5*). The angle is stable at ca. −180° in all

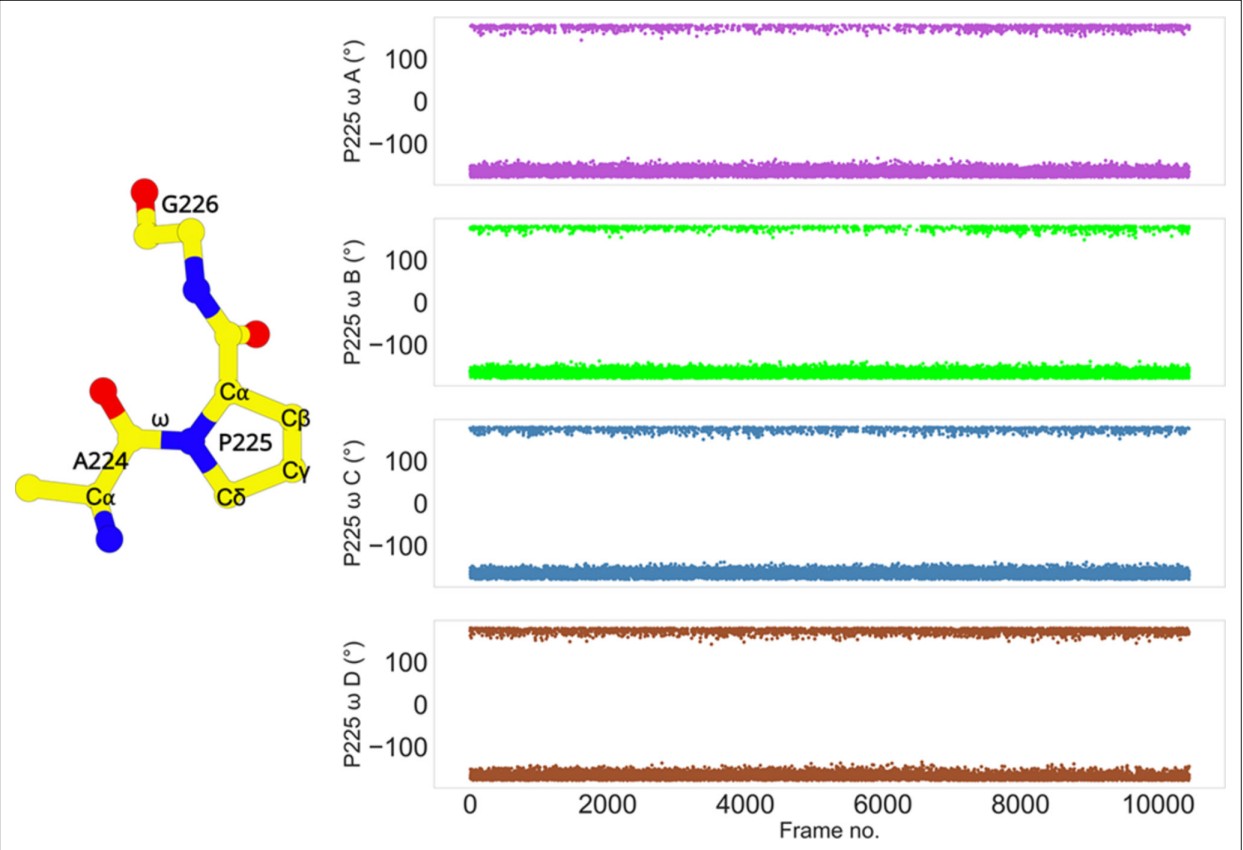

**Figure 5.** Trans conformation of P225. The cis/trans conformation of a peptide bond is determined by the torsion of the $\omega$ angle. The $\omega$ torsion angle is close to 0° in the cis configuration, or ca. 180° in the trans configuration. The $\omega$ angle of P225 in each subunit transitions between 180° and −180°. The stride is 100 frames.

four monomers, while also often transitioning to ca. 180°; thus, confirming the trans configuration of P225. The 'in' and 'out' conformations of P225 identified in different states are regulated by the coupled coordination of π–π stacking (H151-Y227) and the salt bridge (D150c-R236) interactions. These interactions may be the major contributor for stabilizing the trans conformation of P225. Moreover, the transitionary 'in' and 'out' conformational changes of P225 may also be important in for substrate binding.

The side chain of F156 also plays an important role in the ligand-binding process (*Sevaille et al., 2017*; *Spencer et al., 2005*). In fact, all the ligand/substrate-bound forms of L1 MBL structures adopt a closed conformation, presumably due to the tight packing interactions of the side chains in the binding site (*Drawz and Bonomo, 2010*; *Sevaille et al., 2017*; *Spencer et al., 2005*). As observed in each state, the side chain of F156 (*Figure 3*) rotates when transitioning between metastable states. In comparison with the crystal structure, there is an obvious difference in the F156 side-chain dihedral angles in the open, intermediate, and the closed states, suggesting the side-chain rotation of F156 may affect the ligand binding. The functional role of F156 in L1 MBL merits further study.

Several major conformational changes, including the formation and loss of the salt bridge (D150c-R236) and π–π stacking (H151-Y227) interactions, the 'in' and 'out' conformations of P225 existing in trans configuration, the side-chain rotation of F156, have been identified that explain the dynamics of the α3-β7 and β12-α5 loops. These conformational changes of the loops would have significant impact on the topology of the active site, which may further affect the catalytic activity of the L1 MBL. Our findings suggest a plausible two state conformation where β12-α5 loop acts like a lid on the active site. In the open state, the D150c-R236 salt bridge is broken and the π–π stacking interaction between H151 and Y227 is lost. This allows P225 to adopt an 'out' conformation resulting in the loop to move away from the active site. The rearrangement of the residues in the loops increases the volume, which permits the access of the substrate into the active site. Since this is a high energy conformation, the

**Table 2.** Susceptibility results of various β-lactams against L1 variants expressed in the pBC SK (+) vector in *E. coli* DH10B cells.

The reported values are the mode of at least three biological replicates.

| | MIC value (mg/l)* | | | | | |
|---|---|---|---|---|---|---|
| Strain | CAZ | FEP | IMI | MER | TEB | TEB + CS319[†] |
| DH10B pBC SK blaL1a | 256 | 2 | 16 | 16 | 32 | 2 |
| DH10B pBC SK blaL1a_H151V | 2 | 0.03 | 0.125 | 0.03 | ≤0.03 | ≤0.03 |
| DH10B pBC SK blaL1a_D150cV | 256 | 1 | 8 | 8 | 4 | 0.25 |
| DH10B pBC SK blaL1a_P225A | 64 | 0.125 | 0.25 | 2 | 0.125 | ≤0.03 |
| DH10B pBC SK blaL1a_R236A | 256 | 2 | 8 | 8 | 4 | 0.25 |
| DH10B pBC SK blaL1a_Y227A | 64 | 0.25 | 0.25 | 2 | 0.5 | ≤0.03 |

*CAZ, ceftazidime; FEP, cefepime; IMI, imipenem; MER, meropenem; TEB, tebipenem.
[†]CS319 was added to a final concentration of 100 mg/l.

loop is drawn back to a more stable closed conformation, where P225 adopts the 'in' conformation, the π–π stacking and the salt bridge interaction is reformed. The closed conformation is an extremely stable state that is observed in all crystallized conformations of L1 MBL including those with inhibitors (*Crisp et al., 2007*; *Sevaille et al., 2017*; *Spencer et al., 2005*; *Ullah et al., 1998*). These findings in L1 MBL are further supported by experimental studies on two active site loops in NDM-1, IMP-1, and VIM-2 MBLs, which identified the loops to behave like mobile flaps and a direct link exists between the conformation of the loops and the changes in the substrate recognition profile (*Oelschlaeger and Pleiss, 2007*; *Palacios et al., 2019*). Moreover, the flexibility of the loops may also lead to a better positioning of the zinc ion, resulting in an enhanced catalytic profile of the enzyme (*Tomatis et al., 2008*).

## Experimental validation by L1 MBL engineered variants

To study the role of each of the key residues identified in the coordinated dynamics of loops α3-β7 and β12-α5, we engineered the H151V, D150cV, P225A, R236A, and Y227A variants in L1 for susceptibility determinations. The susceptibility of the variants expressed in an isogenic *Escherichia coli* DH10B background was tested against a panel of representative cephalosporins and carbapenems (*Table 2*). We also included a well-studied experimental L1 inhibitor to test the effect of these substitutions in inhibitor binding, L-CS319 (*Hinchliffe et al., 2016*). The H151V substitution had the biggest detrimental effect on the catalytic activity of L1 of all substrates tested, decreasing the minimal inhibitory concentration (MIC) in >6 doubling dilutions. In the case of the P225A and Y227A variants, the effect on the MICs depended upon the substrate. Compared to L1, the MICs for ceftazidime of these variants decreased two-fold dilutions, in contrast to the six-fold dilutions observed for the MIC for imipenem. Lastly, the D150cV and R236A substitution had a minimal effect on the hydrolytic activity of L1, as their MICs values for all substrates were comparable. Addition of the MBL inhibitor L-CS319 further decreased the MIC values, indicating that this compound is able to bind the active site, a fact that is consistent with its small size and the lack of substituents such as those present in the β-lactam antibiotics. The effect of the inhibitor also supports that the enzyme is active in the bacterial periplasm, and that these mutations are not disrupting the protein fold.

## Conclusions

In this report, we have performed enhanced sampling molecular dynamics simulations and built MSM to analyze and understand the dynamics of two flexible loops (α3-β7 and β12-α5 loops) in the L1 enzyme. The deep learning analysis of the MSM cluster centers further resolved the role of individual subunits in the global dynamics of the L1 MBL.

The main focus of this work is to understand the relationship between the loop dynamics and the structural remodeling of the active site. The results presented here suggest that the dynamic properties of the α3-β7 and β12-α5 loops can affect the architecture of the active site. Interactions between

key residues in the loops are primarily responsible for the distinct conformational changes observed. The β12-α5 loop is highly flexible, while the α3-β7 loop is quite stable, due to the direct proximity to the zinc coordination in the active site. The loops adopt 'open' or 'closed' conformations gated by a salt bridge interaction between D150c and R236. The π–π stacking between H151 and Y227 is also crucial. The formation of the stacking interaction together with the salt bridge interaction stabilizes P225 in an 'in' conformation. These conformational changes influence the architecture of the active site and may affect the catalytic activity of L1 MBL. A coordination between the residues in the α3-β7 and β12-α5 loops, leading to the structural remodeling of the active site would be crucial in understanding the conformational changes resulting in the entrance of the substrate and egress of the product from the binding site. It is worth emphasizing that the loop movements occur on the same timescale as the formation of the reaction intermediate (*Carenbauer et al., 2002*, *González et al., 2016*). Therefore, the analyses presented here serve as a starting point for more strategic drug design that capitalizes on these dynamic interactions present in the tetramer. Moreover, our analyses offer the opportunity to search for novel compounds that not only modulate inhibition but will overcome standard approaches that have continued to lead to resistance.

# Materials and methods

## System preparation

The biological functional unit of L1 MBL is a tetramer. The coordinates of the L1 MBL deposited as the PDB entry 1SML are monomeric. The symmetry operators required to generate the tetrameric twofold inverted symmetry are present in the PDB headers. The protein quaternary server (*Henrick and Thornton, 1998*) uses these operators to generate the multimeric biological functional unit of the enzyme, that is a tetramer. The overall structure is shown in (*Figure 1A*), each subunit is in different colors. The PDB file was then protonated at pH 7.0 using proteinprepare module implemented in playmolecule (*Martínez-Rosell et al., 2017*). Since L1 MBL contains zinc ions, the protonated PDB file was adapted to the Zinc Amber force field (ZAFF) format (*Peters et al., 2010*). The system was hydrated using TIP3P water in a cubic waterbox, whose dimension extended to 12 Å from the closest protein atom.

## Adaptive simulations

The ZAFF was used to parameterize the system with electrostatic interactions distances set to 9 Å. Long-range electrostatic interactions were computed using the particle mesh Ewald summation method (*Cerutti et al., 2009*). The system was energy minimized for 1000 iterations of steepest descent and then equilibrated for 5 ns at 1 atmospheric pressure using Berendsen barostat (*Feenstra et al., 1999*). Initial velocities within each simulation were sampled from Boltzmann distribution at temperature of 300 K. Isothermic–isobaric NVT ensemble using a Langevin thermostat with a damping of 0.1 ps$^{-1}$ and hydrogen mass repartitioning scheme to achieve time steps of 4 fs. Multiple short MSM-based adaptively sampled simulations were run using the ACEMD molecular dynamics engine (*Doerr et al., 2016*; *Harvey et al., 2009*). The standard adaptive sampling algorithm performs several rounds of short parallel simulations. To avoid any redundant sampling, the algorithm generates an MSM discretized conformational space and uses the stationary distribution of each state to obtain an estimate of their free energy. It then selects any sampled conformation from a low free energy stable state and respawns a new round of simulations. In this context, the MetricSelfDistance function was set to consider the number of native Cα contacts formed for all residues, which were then used to build the MSMs. The exploration value was 0.01 and goal-scoring function was set to 0.3. For each round, 4 simulations of 50 ns were run in parallel until the cumulative time exceeded 100 μs. The trajectory frames were saved every 0.1 ns. 2090 trajectories were collected with each trajectory counting 500 frames.

## Markov state modeling

PyEMMA v2.5.7 was used to build the MSM (*Scherer et al., 2015*) and carry out kinetic modeling of the tetramer. All 2090 trajectories were loaded into the software. The RMSD and backbone torsions of all residues in α3-β7 (G149-D171) and β12-α5 (A219-H239) loops from all four subunits were selected the input features. Next, the featurized trajectories were read in with a stride of 5 and were

projected onto three independent components (ICs) using TICA. The produced projections can show the maximal autocorrelation for a given lag time (5 ns). The chosen ICs were then clustered into 200 clusters using *k*-means. In this way, each IC was assigned to the nearest cluster center. A lag time of 5 ns was chosen to build an MSM with seven metastable states according to the implied times-cales (ITS) plot (*Figure 2A*). After passing the Chapman–Kolmogorov test within confidence intervals (*Figure 2—figure supplement 1*), the MSM was defined as good. This indicates the model highly agrees with the input data, and it is statistically significant for use. Bayesian MSM was used to build the final model in the system. The net flux pathways between macrostates, starting from state 1, were calculated using Transition Path Theory (TPT) function. These pathways all originate from state 1, as it shows the lowest stationary probability (the highest free energy) in the system. This is why state 1 is a reasonable starting point to illustrate all the relevant kinetic transitions through the full FE landscape. The structural results were selected from each PCCA distribution.

## CVAE-based deep learning

A CVAE-based unsupervised deep learning was implemented to resolve the dynamics of the individual subunits L1 MBL. The CVAE model has been previously used successfully to study many multiple cases including protein folding (*Bhowmik et al., 2018*), enzyme dynamics (*Akere et al., 2020*; *Romero et al., 2019*), COVID-19-related molecular mechanisms (*Cho et al., 2021*), and more specifically β-lac-tamases (*Olehnovics et al., 2021*). The CVAE can identify subtle differences in conformations of 3D structures influenced by various local and global dynamics and is able to cluster them in different microstates (*Yoginath et al., 2019*). The main objective of the CVAE-based deep learning was to cluster the conformations of the α3-β7 and β12-α5 loops sampled in all four subunits independently.

A typical CVAE architecture is a specific deep learning algorithm, built on top of a traditional autoencoder with an added variational approach. In general, an autoencoder has an hourglass-like architecture, where high-dimensional data are fed on one end. As the high-dimensional data passes through the autoencoder, only the essential information is captured. The essential information is then used to reconstruct the high-dimensional data to ensure that there is no loss of information during the compression mechanism of the encoder. The reduced dimension data that is captured is optimized by the variational approach, and is distributed normally, thereby ensuring the efficient utilization of the latent space. The convolutional layers are introduced so that the local and global information is captured in an efficient way from the multilayered complex biomolecular structures.

The distance maps for each monomer were built separately to deconstruct the dynamics of indi-vidual subunits. The pairwise distance maps were built for each Cα atom in the α3-β7 and β12-α5 loops with a cutoff <8 Å. The distance matrices (39 × 39) were stacked in a 3D array. Prior to the training step, the four arrays were concatenated followed by a randomized reordering of the frames. A validation split of 80:20 was defined. A batch size of 300 was used, and the data were reshuffled after each completed epoch. The training was performed for 100 epochs and was assessed by a converged gradient descent. The embeddings were labeled based on P225-Zn distance (6.5 Å), H151-Y227 π–π stacking interaction (4 Å) and D150c-R236 salt bridge interaction (4.5 Å). Identical proce-dure was followed for all four subunits. The python code for the model implemented in the current study was adopted from *Bhowmik et al., 2018*.

## Structural analysis

All trajectories were aligned to their crystal structure conformation using Moleculekit implemented in HTMD tools (*Doerr et al., 2016*). To visualize the structures representing each state, the struc-tures collected from the PCCA distributions were loaded and superimposed in Pymol-mdanalysis (https://github.com/bieniekmateusz/pymol-mdanalysis; *Bieniek, 2022*). The structural analysis was performed using mdtraj (*McGibbon et al., 2015*) and MDAnalysis (*Michaud-Agrawal et al., 2011*). Figures capturing major conformational changes were generated using the Protein Imager (*Tomasello et al., 2020*). All plots were made using the matplotlib libraries (*Hunter, 2007*).

## Susceptibility testing

Variant genes were synthesized into the pBC SK (+) vector by GenScript USA Inc (Piscataway, NJ, USA). Mueller-Hinton (MH) broth microdilution MIC measurements were performed against isogenic trans-formants producing L1a and five variants of L1a in pBC SK (+) according to the Clinical and Laboratory

Standards Institute (CLSI) protocol (*Wayne, 2020*). L-CS319 was tested at a constant 100 µg/ml in combination with various concentrations of tebipenem. The MICs are reported as the concentrations at which bacterial growth was no longer observed. All MIC measurements were performed at least three times and the mode value has been reported.

## Acknowledgements

We would like to thank Dr. Graciela Mahler for providing L-CS319. RAB and MFM received grant support from US CDC Prevention Epicenters Program under the awards number U54CK000603. RAB acknowledges a grant from the National Institutes of Health United States under the award number R01AI100560. RAB has received research support from Shionogi, VenatoRx, Wockhardt, Merck, and Entasis in the past 2 years. The funders have no role in the study design, data collection, and interpretation, of the decision to submit the work for publication. This material is based upon work supported by the U.S. Department of Energy, Office of Science, Office of Advanced Scientific Computing Research, under contract number DEAC05- 00OR22725. This research is sponsored in part by the Laboratory Directed Research and Development Program of Oak Ridge National Laboratory, managed by UT-Battelle, LLC, for the U.S. Department of Energy. This research used resources of the Oak Ridge Leadership Computing Facility at the Oak Ridge National Laboratory, which is supported by the Office of Science of the U.S. Department of Energy under Contract no. DE-AC05-00OR22725. The US government retains and the publisher, by accepting the article for publication, acknowledges that the US government retains a nonexclusive, paid-up, irrevocable, worldwide license to publish or reproduce the published form of this manuscript, or allow others to do so, for US government purposes. DOE will provide public access to these results of federally sponsored research in accordance with the DOE Public Access Plan (http://energy.gov/downloads/doe-public-access-plan).

## Additional information

### Competing interests

Shozeb Haider: Reviewing editor, eLife. The other authors declare that no competing interests exist.

### Funding

| Funder | Grant reference number | Author |
| --- | --- | --- |
| National Institutes of Health | R01AI100560 | Robert A Bonomo |
| US CDC Prevention Epicenters Program | U54CK000603 | Robert A Bonomo Maria F Mojica |

The funders had no role in study design, data collection, and interpretation, or the decision to submit the work for publication.

### Author contributions

Zhuoran Zhao, Xiayu Shen, Formal analysis, Investigation, Visualization, Writing – original draft; Shuang Chen, Formal analysis, Visualization; Jing Gu, Haun Wang, Investigation, Visualization; Maria F Mojica, Formal analysis, Investigation, Writing – review and editing; Moumita Samanta, Software, Investigation, Methodology; Debsindhu Bhowmik, Software, Validation, Visualization, Methodology; Alejandro J Vila, Supervision, Writing – review and editing; Robert A Bonomo, Supervision, Writing – review and editing, Funding acquisition; Shozeb Haider, Conceptualization, Data curation, Formal analysis, Supervision, Funding acquisition, Validation, Investigation, Visualization, Methodology, Project administration, Writing – review and editing

### Author ORCIDs

Maria F Mojica http://orcid.org/0000-0002-1380-9824
Debsindhu Bhowmik http://orcid.org/0000-0001-7770-9091
Alejandro J Vila http://orcid.org/0000-0002-7978-3233
Shozeb Haider http://orcid.org/0000-0003-2650-2925

**Decision letter and Author response**
Decision letter https://doi.org/10.7554/eLife.83928.sa1
Author response https://doi.org/10.7554/eLife.83928.sa2

## Additional files

**Supplementary files**
• MDAR checklist

**Data availability**
The simulation data including all trajectories and the corresponding structure files can be downloaded from the https://doi.org/10.57760/sciencedb.06948.

The following dataset was generated:

| Author(s) | Year | Dataset title | Dataset URL | Database and Identifier |
|---|---|---|---|---|
| Haider S | 2023 | A dataset of L1 MBL simulation | https://doi.org/10.57760/sciencedb.06948 | Science Data Bank, 10.57760/sciencedb.06948 |

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
