## [Editor Report]

In this useful study, the authors utilize state-of-the-art computational methods complemented with some experimental validation to investigate the dynamics of flexible loops of the L1 Metallo-β-lactamase enzyme, resulting in a better understanding of the various conformational states useful for the rational design of superior β-lactamase inhibitors/antibiotics. The evidence supporting the claims is solid, and the work will be of interest to computational, experimental biologists, and drug designers working on antibiotic resistance.

---

## [Decision Letter]

**Decision letter after peer review:**

Thank you for submitting your article "Gating interactions steer loop conformational changes in the active site of the L1 metallo β-lactamase" for consideration by *eLife*. Your article has been reviewed by 3 peer reviewers, and the evaluation has been overseen by a Reviewing Editor and Volker Dötsch as the Senior Editor. The following individual involved in the review of your submission has agreed to reveal their identity: Davide Provasi (Reviewer #2).

Essential revisions:

Please address the concerns of three reviewers, which are appended below.

*Reviewer #1 (Recommendations for the authors):*

1. Authors mention "D150c" in several parts of the paper and in figure labels. Authors should clarify this nomenclature for a broader audience.

2. Figure 2A needs a legend.

3. On line 237, the authors state residue 150 is S, whereas throughout the rest of the paper they discuss it as D150. Please clarify and correct.

4. Lines 303-308 have 39 mentioned in the text -please correct.

5. In the experimental testing step – a larger diversity of residues being tested would enable a broader understanding of the active site. This study would be further enhanced by double mutant variants to enable the authors to understand whether the bond type is important for the active site or if different bond types can exist to stabilize these loops (H-Bonding instead of pi-pi stacking as an example). This enhances the claim of the paper being useful for drug design. Further, there are several amino acids mentioned as being important at the beginning of the manuscript, yet we do not see their trajectories later in the manuscript. These would enhance the authors' claims as well.

6. How were the 22 residues identified and which are they? Understanding this is important for manuscript flow and would improve the narrative of the study. Further, data split results, and reducing input data matrix below 22 residues are all factors that should be tested for the validity of study results.

*Reviewer #2 (Recommendations for the authors):*

As outlined in the public review, I think that the combination of CVAE and dihedral featurisation is confusing. If sophisticated CVAE embedding is necessary to resolve the relevant dynamical features, then it should be used instead of the dihedrals+tICA to build the MSM. Microstates for the MSM analysis could be easily defined based on k-means clustering of the 14-dimensional CVAE latent space, providing a unified description of the dynamics. If, on the other hand, the dihedral+tICA featurisation is enough to obtain a descriptive and converged model, what is the additional insight offered by the deep learning approach?

Another concern is that presently, all results from the CVAE are analysed and presented after a further projection with tSNE. What is the effect of the compression to 2 dimensions from the original dimensions of the VAE latent space?

The description of the methods is unclear or insufficient on several points:

1) System preparation. The system preparation strategies should be justified better. The authors simulate a homo-tetramer, building the complex from the monomer crystal structure. This choice should be justified, and the result of the tetramer modeling validated. Why should we trust the complex structure if there are no experimental structures of the tetramer?

2) Adaptive Simulations. the adaptive simulation strategy is not described. The authors just mention that "Multiple short Markov State Model (MSM)-based adaptively sampled simulations were run". How? What were the criteria to build the MSM used to re-spawn adaptively the simulations? Was it the same as the one used for the analysis?

3) MSM. It should be made more clear in the methods that the dynamics of each of the four tetrameric units are used by considering the four units to be independent and featuring the dynamics of each unit, and not of the full tetramer. This is a good strategy to increase the available data for kinetic modeling, but it means that no conclusions can be extracted from cooperative effects across subunits. The CK test results should probably be reported as figure supplements.

For the results and discussions section:

1) confidence intervals should be reported for timescales (Figure 2A), mean first-passage times (Figure 2C), probabilities, and free-energies (Figure 2D). Error bars should be reported for the experimental data too (the authors mention triplicate experiments but only report one value per condition).

2) The description of the MSM results can be greatly improved and streamlined. For instance, the description of the structural features of the 7 macrostates (paragraph "The α3-β7 and β12-α5 loops exist in open, intermediate and closed states") should come before the description of the most probable paths connecting the different states. Furthermore, to clarify and facilitate the identification of the structural features of the 7 macrostates, the distributions (or averages and percentiles) of the key distances mentioned in the results (e.g., D150c-R236, H151-Y227), etc. should be reported for each state, similar to what has been done in Figure 3D.

3) "States 3, 5, and 6 are sub-states of the closed conformation, with differences in the conformations of loops joining β-sheets 10 and 11, far away from the active site". It's unclear how the macrostates, which were defined based on features describing only the α3-β7 and β12-α5 loops, could resolve structural features unrelated to these two regions.

*Reviewer #3 (Recommendations for the authors):*

Although a very thoughtful and detailed MD study, several additional inputs would further strengthen the study. They are described below:

1) Using orthologs, it will be good to provide the percentage of MBLs (e.g., say at 85% sequence identity) that have elongated alpha3-beta7 and beta12-alpha5 types of loops. AlphaFold2 structures of such sequences can further facilitate loop content estimation. This will enhance the relevance of the study, i.e., applicable to not just one (Stenotrophomonas maltophilia MBL L1) but to MBLs of other pathogens. If the percent of MBLs with elongated loops is say < 10%, should this study be carried out? In that case (i.e., if the elongated loops in MBLs are uncommon.), the threat from Stenotrophomonas maltophilia will need to be emphasized further for the significance of the study.

2) As a nonpolar surface patch is responsible for MBL L1 multimerization, can alphafold2 also help in estimating the number of multimeric MBLs that are possible based on the non-polar surface patches of alphafold2 modeled MBL proteins? That is, an estimate of the proportion of expected multimeric MBLs with elongated loops would help enhance the relevance of the study to a wider collection of MBLs.

3) In describing the conformational landscape of the loops, is mutagenesis the right experiment to capture the essence of the interacting loop residues? To illustrate this point, let there be two situations: (a) only one conformation for a loop which is held in place by a salt bridge, (b) two conformations for the loop, (1/3rd time in conformation-i and 2/3rd time in conformation-j) where conformation-i is held in place by the salt bridge. Can mutagenesis of the salt bridge residues distinguish scenario (a) from that of (b)?

4) Alternatively, let us say the following scenarios: (a) only one conformation for a loop (i.e., an open conformation) with no specific interaction, (b) two conformations for the loop (1/3rd time in conformation-i and 2/3rd time in conformation-j) where conformation-i is held in place by the salt bridge. Can mutagenesis of the salt bridge residues distinguish scenario (a) from that of (b)? It is better to describe the scenarios as in (3) or (4) or some others that are being probed and how mutagenesis captures or distinguishes the different scenarios. From Figure 1B, the impression is that scenarios (in (3) above) are likely the case and it is obvious that mutagenesis of the salt bridge residues will have an impact even if the conformational states were not identified by the current study. [In Figure 1, label '(C)' should be in place of label '(B)' in the legend]

5) As the ultimate goal of such a study is to identify better inhibitors, would the following investigation be a better way to utilize the conformational states of the loops? For example, let us say there are 3 loop conformations, i, j, and k that are captured by the enhanced MD sampling and MSM methods. Can each conformation (i.e., i, j, and k) be utilized separately for virtual screening of small molecules (e.g., using Schrodinger such as in ) to demonstrate that searching with each conformation independently (i.e., as three pocket volumes were observed in Figure 3D) provides higher success in the virtual screen than that of the single conformation of the experimental structure?

6) Authors mention specific interactions of the closed state (e.g., salt bridge R236-D150c, pi-pi stacking Y227-H151, and Q310 sidechain-mainchain hydrogen bond). These interactions are lost in the open state. How does the loss of the 3 interaction energies of the closed state get compensated in the open state? What is the population density of each of the open, intermediate, and closed states? Can the difference (i.e., loss or gain) in the energies due to the 3 interactions (R236-D150c, Y227-H151, and Q310) help explain the population densities of each state?

7) The authors mention Y227-H151 pi-pi stacking. The surface-exposed H151 residue can be protonated as well. That is, Y227-H151 could be a cation-pi too. Can the protonation status of H151 further influence the observed outcome?

---

## [Author Response]

Reviewer #1 (Recommendations for the authors):1. Authors mention "D150c" in several parts of the paper and in figure labels. Authors should clarify this nomenclature for a broader audience.

We appreciate the reviewer’s concern. We have added supplementary figure-1 and a note describing the nomenclature and corresponding numbering used throughout the manuscript.

2. Figure 2A needs a legend.

A legend has been added to Figure 2A. It now reads –

“(A) Implied timescales (ITS) plot. The ITS plot describes the convergence behaviour of the implied timescales associated with the processes at the lag time of 5 ns.”

3. On line 237, the authors state residue 150 is S, whereas throughout the rest of the paper they discuss it as D150. Please clarify and correct.

The numbering in class B β-lactamases have historically been very complex (doi: 10.1128/AAC.48.7.2347-2349.2004). In some cases, like L1 MBL, there are unconventional numbers to accommodate extra residues in loops like those described in this manuscript. For example, proceeding residue 150 is 150a, 150b 150c and 150d. Residue 150a is a Serine, while residue 150c is an Aspartic acid. We have now added Figure 1 —figure supplement 2 that addresses the nomenclature and clarifies the numbering of L1 MBL used in this manuscript.

4. Lines 303-308 have 39 mentioned in the text -please correct.

This was a typo and has now been removed.

5. In the experimental testing step – a larger diversity of residues being tested would enable a broader understanding of the active site. This study would be further enhanced by double mutant variants to enable the authors to understand whether the bond type is important for the active site or if different bond types can exist to stabilize these loops (H-Bonding instead of pi-pi stacking as an example). This enhances the claim of the paper being useful for drug design. Further, there are several amino acids mentioned as being important at the beginning of the manuscript, yet we do not see their trajectories later in the manuscript. These would enhance the authors' claims as well.

The reviewer is correct in pointing out that expanding the cohort of amino acids would enhance the broader understanding of the active site. Based on various published crystal structures, many of these residues in the active site that are in direct contact with the zinc ions or in the second shell of interactions have been mutated and their effects have been reported (https://journals.asm.org/doi/10.1128/mBio.00405-19). Our choice of amino acids was based entirely on the dynamic roles extracted from molecular dynamics simulations. We agree that, indeed, there would be additional residues that would play an important role in the stabilisation of these loops (directly or indirectly). We are in the process of identifying amino acids, studying their dynamic roles and experimentally testing them for a follow up study. Similarly, we designed the single mutants with the intention to disrupt specific residue-residue interactions that could affect the stability of the loops. The experimental data obtained so far supports out hypothesis regarding the role of those residues in the stabilization of the loops and their effect on substrate binding/catalysis. More experiments would need to be conducted to assess the specific effect of each substitution in protein stability, zinc affinity and substrate binding (Km) or catalysis (Kcat) that explains each specific phenotype. Once we have established the effect of each individual substitution, we could design double mutants to evaluate more possible scenarios that would ultimately deepen our knowledge of this enzyme.

6. How were the 22 residues identified and which are they? Understanding this is important for manuscript flow and would improve the narrative of the study. Further, data split results, and reducing input data matrix below 22 residues are all factors that should be tested for the validity of study results.

This is an excellent point raised by the reviewer. The loop residues were selected between two structural elements- α3-β7 (residues 149-171) and β12-α5 (residues 219-239) (Table 1). All 39 residues from α3-β7 and β12-α5 loops were used to build our MSMs. However, since the loops are quite long, we truncated our selection based on the dynamics identified from the MSMs. This is how we selected 22 residues and built our 22x22 matrix for the CVAE analysis.

However, we agree with the reviewer on the importance of understanding the selection of these residues and the narrative of the study. Therefore, we have expanded the selection to include ALL residues in the α3-β7 and β12-α5 loops. The residue boundaries have been mentioned on line 239-240. A list of residues included in the loops, that make up the matrix are also listed in Table 1.

As suggested by the reviewer, we further reanalysed the data for three different sets of matrices besides the 22x22 matrix previously reported. First, we included ALL residues of two loops in a 39x39 matrix (data split 80:20); followed by 28x28 (residues G149-D161 and A219-P238; data split 80:20 and 70:30) and 17x17 (residues D150b-F156 and S223-R236; data split 80:20 and 70:30) matrices. The results from the new runs have been illustrated in Author response images 1-4. Since the results between 39x39, 28x28 and 22x22 matrices are similar, we have now included the results from the 39x39 matrix in Figure 4. Figure 4 —figure supplement 1 has also been changed to correspond with the results from the 39x39 matrix.

**Author response image 1. sa2fig1:** CVAE based deep learning analysis from a 28x28 matrix with 80 (training):20 (validation) data split. (a) The training and validation loss plot as assessed over consecutive epochs at various reduced dimensions. (b) Validation loss during CVAE implementation is plotted at different latent dimensions for determining optimum values of the low dimension. The loss was calculated over 100 epochs from dimensions 3-30. (c) The convergence of training and validation loss from dimension 29, as assessed over 100 epochs. (d) Comparison between original input data and reconstructed data (decoded) are illustrated to ensure no loss of essential information during the compression and reconstruction process. (e) Subunit representation in the tsne space. The (f)salt bridge interaction between D150c-R236, (g) H151-Y227 p-p stacking interaction and (h) distance between P225 and the centroid of Zn atoms.

**Author response image 2. sa2fig2:** CVAE based deep learning analysis from a 28x28 matrix with 70 (training):30 (validation) data split. (**a**) The training and validation loss plot as assessed over consecutive epochs at various reduced dimensions. (**b**) Validation loss during CVAE implementation is plotted at different latent dimensions for determining optimum values of the low dimension. The loss was calculated over 100 epochs from dimensions 3-30. (**c**) The convergence of training and validation loss from dimension 28, as assessed over 100 epochs. (**d**) Comparison between original input data and reconstructed data (decoded) are illustrated to ensure no loss of essential information during the compression and reconstruction process. (**e**) Subunit representation in the tsne space. The (**f**) salt bridge interaction between D150c-R236, (**g**) H151-Y227 p-p stacking interaction and (**h**) distance between P225 and the centroid of Zn atoms.

**Author response image 3. sa2fig3:** CVAE based deep learning analysis from a 17x17 matrix with 80 (training):20 (validation) data split. (**a**) The training and validation loss plot as assessed over consecutive epochs at various reduced dimensions. (**b**) Validation loss during CVAE implementation is plotted at different latent dimensions for determining optimum values of the low dimension. The loss was calculated over 100 epochs from dimensions 3-30. (**c**) The convergence of training and validation loss from dimension 19, as assessed over 100 epochs. (**d**) Comparison between original input data and reconstructed data (decoded) are illustrated to ensure no loss of essential information during the compression and reconstruction process. (**e**) Subunit representation in the tsne space. The (**f**) salt bridge interaction between D150c-R236, (**g**) H151-Y227 p-p stacking interaction and (**h**) distance between P225 and the centroid of Zn atoms.

**Author response image 4. sa2fig4:** CVAE based deep learning analysis from a 17x17 matrix with 70 (training): 30 (validation) data split. (**a**) The training and validation loss plot as assessed over consecutive epochs at various reduced dimensions. (**b**) Validation loss during CVAE implementation is plotted at different latent dimensions for determining optimum values of the low dimension. The loss was calculated over 100 epochs from dimensions 3-30. (**c**) The convergence of training and validation loss from dimension 29, as assessed over 100 epochs. (**d**) Comparison between original input data and reconstructed data (decoded) are illustrated to ensure no loss of essential information during the compression and reconstruction process. (**e**) Subunit representation in the tsne space. The (**f**) salt bridge interaction between D150c-R236, (**g**) H151-Y227 p-p stacking interaction and (**h**) distance between P225 and the centroid of Zn atoms.

Reviewer #2 (Recommendations for the authors):As outlined in the public review, I think that the combination of CVAE and dihedral featurisation is confusing. If sophisticated CVAE embedding is necessary to resolve the relevant dynamical features, then it should be used instead of the dihedrals+tICA to build the MSM. Microstates for the MSM analysis could be easily defined based on k-means clustering of the 14-dimensional CVAE latent space, providing a unified description of the dynamics. If, on the other hand, the dihedral+tICA featurisation is enough to obtain a descriptive and converged model, what is the additional insight offered by the deep learning approach?

We appreciate the reviewer’s comment that the use of dihedral featurisation and then the subsequent use of CVAE might seem confusing. The use of such a protocol is not unprecedented and has been published elsewhere (https://doi.org/10.1371/journal.pcbi.1006801). To place this in the context of our study, we were able to build Markovian models using dihedrals+tICA as our data reduction method (using the backbone atoms and RMSD of loops as input features) and subsequently used them for structural comparisons. The MSMs were generated from trajectories of the tetrameric L1. MSM was preferred here since we wanted to study the transitions between metastable states.

We then used CVAE as a second method to analyse the dynamics of each of the four individual subunits independently. The CVAE method was sensitive enough to identify subtle conformational differences between the metastable states. The use of CVAE allowed us to simultaneously observe various distances, and identify subunits that contributed to these individual conformations.

To summarise, by using dihedral+tICA we were able to carry out kinetic modelling of the tetramer, while CVAE helped us study the dynamics of individual subunits.

We also agree with the reviewer that one of the routes that we could have taken was to use CVAE clustered data and integrate it to generate our Markov state models. However, we went with tICA instead in this instance. We have therefore added a sentence in the methods section and discussion emphasising our rationale on the use of CVAE.

Another concern is that presently, all results from the CVAE are analysed and presented after a further projection with tSNE. What is the effect of the compression to 2 dimensions from the original dimensions of the VAE latent space?

CVAE results are analysed and further projected with tSNE with the purpose of easier visualisation in lower dimensions (such as 2D or 3D) that are more natural for the human eye to follow. That means finding an easier visualisation scheme that facilitates efficient non-linear mapping. But for this non-linear mapping, one of the basic obstacles to overcome is to find an effective solution to the crowding problem i.e. finding how to segregate neighbouring modestly spaced data points into lower dimension while simultaneously maintaining the separation among the clusters. tSNE achieves this by converting these distances to probability scores in the projected lower dimensions while simultaneously allowing to interpret original complex relation and maintaining global and local structure of the data.

The description of the methods is unclear or insufficient on several points:1) System preparation. The system preparation strategies should be justified better. The authors simulate a homo-tetramer, building the complex from the monomer crystal structure. This choice should be justified, and the result of the tetramer modeling validated. Why should we trust the complex structure if there are no experimental structures of the tetramer?

We would like to clarify that we have not modelled the tetramer, but rather generated the biological functional unit using the symmetry related operators via the PQS server. There are several structures of L1 MBL present in the PDB. Some are deposited as dimers (PDB entry 5DPX) and others as tetramers (PDB entry 2AIO). Our choice of structure was based on the fact that PDB entry 1SML was an apo structure unlike others which are in complex with small molecules. The overall RMSD between 1SML and 5DPX structures over 1064 Ca residues is 0.77 Å.

We have added further details to the system preparation section. The section now reads as follows:

“The biological functional unit of L1 metallo-b-lactamase is a tetramer. The coordinates of the L1 MBL deposited in the PDB entry 1SML are monomeric. The symmetry operators required to generate the tetrameric two-fold inverted symmetry are present in the PDB headers. The protein quaternary server (Henrick, 1998) server uses these operators to generate the multimeric biological functional unit of the enzyme. i.e. a tetramer. The overall structure is shown in (Figure 1A), each subunit is in different colors. The PDB file was then protonated at pH 7.0 using proteinprepare module implemented in playmolecule (Martínez-Rosell et al., 2017). Since L1 MBL contains zinc ions, the protonated PDB file was adapted to the Zinc Amber force field (ZAFF) format (Peters et al., 2010). The system was hydrated using TIP3P water in a cubic waterbox, whose dimension extended to 12 Å from the closest protein atom.”

2) Adaptive Simulations. the adaptive simulation strategy is not described. The authors just mention that "Multiple short Markov State Model (MSM)-based adaptively sampled simulations were run". How? What were the criteria to build the MSM used to re-spawn adaptively the simulations? Was it the same as the one used for the analysis?

This has been added to methods section and now reads:

“Multiple short Markov state model based adaptively sampled simulations were run using the ACEMD molecular dynamics engine. The standard adaptive sampling algorithm performs several rounds of short parallel simulations. To avoid any redundant sampling, the algorithm generates an MSM’s discretized conformational space and uses the stationary distribution of each state to obtain an estimate of their free energy. It then selects any sampled conformation from a low free energy stable state and respawns a new round of simulations. In this context, the MetricSelfDistance function was set to consider the number of native Ca contacts formed from all residues, which were then used to build the MSMs. The exploration value was 0.01 and goal-scoring function was set to 0.3. For each round, 4 simulations of 50 ns were run in parallel until the cumulative time exceeded 100 µs. The trajectory frames were saved every 0.1 ns. 2090 trajectories were collected with each trajectory counting 500 frames.”

3) MSM. It should be made more clear in the methods that the dynamics of each of the four tetrameric units are used by considering the four units to be independent and featuring the dynamics of each unit, and not of the full tetramer. This is a good strategy to increase the available data for kinetic modeling, but it means that no conclusions can be extracted from cooperative effects across subunits. The CK test results should probably be reported as figure supplements.

We like to point out that the dynamics of all four tetrameric subunits was used to build the MSMs. It was in CVAE that the dynamics of each tetrameric unit was considered independently. This has now been clarified added to the methods section. A CK plot has been added as Figure 2 —figure supplement 1 as advised by the reviewer.

For the results and discussions section:1) confidence intervals should be reported for timescales (Figure 2A), mean first-passage times (Figure 2C), probabilities, and free-energies (Figure 2D). Error bars should be reported for the experimental data too (the authors mention triplicate experiments but only report one value per condition).

We have now added confidence intervals for the ITS plot and mean first-passage times. Regarding the MIC data, the value reported is the mode out of at least 3 biological replicates. This has now been mentioned in the methods. To emphasise this, we have also added this in the table legend.

2) The description of the MSM results can be greatly improved and streamlined. For instance, the description of the structural features of the 7 macrostates (paragraph "The α3-β7 and β12-α5 loops exist in open, intermediate and closed states") should come before the description of the most probable paths connecting the different states. Furthermore, to clarify and facilitate the identification of the structural features of the 7 macrostates, the distributions (or averages and percentiles) of the key distances mentioned in the results (e.g., D150c-R236, H151-Y227), etc. should be reported for each state, similar to what has been done in Figure 3D.

As suggested by the reviewer, the paragraph "The α3-β7 and β12-α5 loops exist in open, intermediate and closed states" has been moved before the description of the probable paths. The violin plots of the interactions from each metastable state have been added as Figure 3 —figure supplement 2-5.

3) "States 3, 5, and 6 are sub-states of the closed conformation, with differences in the conformations of loops joining β-sheets 10 and 11, far away from the active site". It's unclear how the macrostates, which were defined based on features describing only the α3-β7 and β12-α5 loops, could resolve structural features unrelated to these two regions.

The reviewer is correct in pointing this out. The b-sheets are at the distal ends of the active site and directly connected to loop β12-α5. We have reworded the sentence, which now reads:

“States 3, 5 and 6 are sub-states of the closed conformation. In addition to the slight differences in the conformations of the a3-b7 and b12-a5 loops, minor changes are also observed in the conformations of the adjacent loops at the distal end of the active site.”

Reviewer #3 (Recommendations for the authors):Although a very thoughtful and detailed MD study, several additional inputs would further strengthen the study. They are described below:1) Using orthologs, it will be good to provide the percentage of MBLs (e.g., say at 85% sequence identity) that have elongated alpha3-beta7 and beta12-alpha5 types of loops. AlphaFold2 structures of such sequences can further facilitate loop content estimation. This will enhance the relevance of the study, i.e., applicable to not just one (Stenotrophomonas maltophilia MBL L1) but to MBLs of other pathogens. If the percent of MBLs with elongated loops is say < 10%, should this study be carried out? In that case (i.e., if the elongated loops in MBLs are uncommon.), the threat from Stenotrophomonas maltophilia will need to be emphasized further for the significance of the study.

The a3-b7 and b12-a5 elongated loops are a structural feature of all B3 MBLs. Subclass B3 groups enzymes from *Stenotrophomonas maltophilia*, FES^-1^ from *Legionella gormanii*, SMB-1 from *Serratia marcescens* and GOB-type enzymes from *Elizabethkingia meningoseptica*, BJP-1 from *Bradyrhizobium diazoefficiens*. *B. diazoefficiens* is a bacterium of considerable agricultural importance, as it is the most widely used species in commercial inoculants. This symbiont species of many plants induces the formation of nodules in the roots, which are the specialised structures where N2 fixation takes place. Furthermore, in addition to the highly diverse L1 b-lactamase family members (>73 clinical isolates from *Stenotrophomonas maltophilia*), POM1/2 (*Pseudomonas otitidis*), PAM1/2/3 (from *Pseudomona alcaligenes*, *Pseudomonas tohonis*), DHT2-1 (uncultured bacterium) are other b-lactamases that share ~52-64% sequence identity with α3-β7 and β12-α5 loops. The bla_POM_ gene is the first example of an MBL gene identified in pathogenic *Pseudomonas* species. However, little is known about the clinical importance of these MBLs mainly because of the difficulties in identifying and distinguishing these bacteria from the other closely related *Pseudomonas* species. With no structure available for these b-lactamases, the L1 MBL acts like a model to understand the function of these enzymes.

**Author response image 5. sa2fig5:** 

Author response image 5 illustrates superimposition of several B3 enzymes: L1 in green (PDB 1SML), SMB-1 in cyan (PDB 3VQZ), FEZ-1 in magenta (PDB 1JTZ) and BJP-1 in yellow (PDB 3LVZ), all of them displaying the elongated active site loops.

2) As a nonpolar surface patch is responsible for MBL L1 multimerization, can alphafold2 also help in estimating the number of multimeric MBLs that are possible based on the non-polar surface patches of alphafold2 modeled MBL proteins? That is, an estimate of the proportion of expected multimeric MBLs with elongated loops would help enhance the relevance of the study to a wider collection of MBLs.

L1 MBL is unique in its structural architecture with the presence of the non-polar N-terminal loop that is essential for tetramerization. In addition to L1 MBL, POM1/2, PAM1/2/3 and DHT2-1 MBLs also possess this N-terminal non-polar loop. Therefore, a multimer generated for the MBLs with homologous non-polar loop via AlphaFold would also be predicted as a tetramer, since most of the MSA information will come from the closest homolog – the L1 MBL. However, there has been no experimental verification that these MBLs are oligomeric. Furthermore, DeepMind team and others have shown that AlphaFold predictions vary substantially in their global and local agreement with only 40% of residues in the human proteome modelled with high confidence (iScience 25, 104496 (2022), PLOS Computational Biology 18, e1009818 (2022)). While some AlphaFold predictions are astonishingly accurate, many predictions are incompatible with experimental data from corresponding crystal structures (https://www.biorxiv.org/content/10.1101/2022.11.21.517405v1). To summarise, L1 MBL can act like a template to model individual subunits, however predicting multimerization in absence of experimental data or verification would at best be speculative.

3) In describing the conformational landscape of the loops, is mutagenesis the right experiment to capture the essence of the interacting loop residues? To illustrate this point, let there be two situations: (a) only one conformation for a loop which is held in place by a salt bridge, (b) two conformations for the loop, (1/3rd time in conformation-i and 2/3rd time in conformation-j) where conformation-i is held in place by the salt bridge. Can mutagenesis of the salt bridge residues distinguish scenario (a) from that of (b)?

We believe mutagenesis is one of the best experiments to describe the essence of the interacting loop residues. From the scenarios that the reviewer presents, there would be a large change in (a) and to a lesser extent in (b). A loss of the salt bridge in (b) would not shift the equilibrium resulting from a loss of conformation-i to conformation-j.

4) Alternatively, let us say the following scenarios: (a) only one conformation for a loop (i.e., an open conformation) with no specific interaction, (b) two conformations for the loop (1/3rd time in conformation-i and 2/3rd time in conformation-j) where conformation-i is held in place by the salt bridge. Can mutagenesis of the salt bridge residues distinguish scenario (a) from that of (b)? It is better to describe the scenarios as in (3) or (4) or some others that are being probed and how mutagenesis captures or distinguishes the different scenarios. From Figure 1B, the impression is that scenarios (in (3) above) are likely the case and it is obvious that mutagenesis of the salt bridge residues will have an impact even if the conformational states were not identified by the current study. [In Figure 1, label '(C)' should be in place of label '(B)' in the legend]

We agree with the reviewer that it is better to describe the scenarios as they are being probed. There have been numerous other studies that carry out alanine scanning of multiple residues and then describe the effects of the mutations. However, in our context, we have rationalised the important residues based on the dynamics of the loops and then subsequently mutated them to confirm the predictions from the simulations. The typo has been corrected.

5) As the ultimate goal of such a study is to identify better inhibitors, would the following investigation be a better way to utilize the conformational states of the loops? For example, let us say there are 3 loop conformations, i, j, and k that are captured by the enhanced MD sampling and MSM methods. Can each conformation (i.e., i, j, and k) be utilized separately for virtual screening of small molecules (e.g., using Schrodinger such as in ) to demonstrate that searching with each conformation independently (i.e., as three pocket volumes were observed in Figure 3D) provides higher success in the virtual screen than that of the single conformation of the experimental structure?

Like the reviewer has suggested, it is a possibility that we also envisage. For this very reason we conclude our study with the sentence. “Therefore, the analyses presented here serves as a starting point for more strategic drug design that capitalizes on these dynamics interactions present in the tetramer. Moreover, our analyses offers the opportunity to search for novel compounds that not only modulate inhibition but will overcome standard approaches that have continued to lead to resistance.”

6) Authors mention specific interactions of the closed state (e.g., salt bridge R236-D150c, pi-pi stacking Y227-H151, and Q310 sidechain-mainchain hydrogen bond). These interactions are lost in the open state. How does the loss of the 3 interaction energies of the closed state get compensated in the open state? What is the population density of each of the open, intermediate, and closed states? Can the difference (i.e., loss or gain) in the energies due to the 3 interactions (R236-D150c, Y227-H151, and Q310) help explain the population densities of each state?

Based on the reviewer’s suggestion we calculated the pairwise interaction energies between residues. We observe that even in the open state, when there is a considerable change in energy; at no point we observe unfavourable energies between the residues. Therefore, in the open state, the loops behave like a high-tension spring, which is energetically pushed to a more relaxed (stable) and preferred closed conformation. It is therefore not surprising that loops in all crystal structures are in closed conformation.

**Author response image 6. sa2fig6:** 

Furthermore, we calculated the population density profiles of the described interactions in the open, intermediate and closed states. We believe that the difference in energies due to 3 interactions would not give an accurate representation of the population densities of each state as other interactions from the loops and neighbouring structural elements would also be contributing towards the total energetics of the open, intermediate and closed states.

**Author response image 7. sa2fig7:** 

7) The authors mention Y227-H151 pi-pi stacking. The surface-exposed H151 residue can be protonated as well. That is, Y227-H151 could be a cation-pi too. Can the protonation status of H151 further influence the observed outcome?

That is a possibility, but we have no way to verify it as we are carrying out a homogenous pH simulation.